# Chest X-ray Bone Suppression for Improving Classification of Tuberculosis-Consistent Findings

**DOI:** 10.3390/diagnostics11050840

**Published:** 2021-05-07

**Authors:** Sivaramakrishnan Rajaraman, Ghada Zamzmi, Les Folio, Philip Alderson, Sameer Antani

**Affiliations:** 1National Library of Medicine, National Institutes of Health, Bethesda, MD 20814, USA; ghadazamzmi.alzamzmi@nih.gov (G.Z.); sameer.antani@nih.gov (S.A.); 2Clinical Center, Department of Radiology and Imaging Sciences, National Institutes of Health, Bethesda, MD 20814, USA; les.folio@nih.gov; 3School of Medicine, Saint Louis University, St. Louis, MO 63103, USA; philip.alderson@health.slu.edu

**Keywords:** deep learning, bone suppression, tuberculosis, convolutional neural networks, classification, statistical analysis, interpretation, chest X-rays

## Abstract

Chest X-rays (CXRs) are the most commonly performed diagnostic examination to detect cardiopulmonary abnormalities. However, the presence of bony structures such as ribs and clavicles can obscure subtle abnormalities, resulting in diagnostic errors. This study aims to build a deep learning (DL)-based bone suppression model that identifies and removes these occluding bony structures in frontal CXRs to assist in reducing errors in radiological interpretation, including DL workflows, related to detecting manifestations consistent with tuberculosis (TB). Several bone suppression models with various deep architectures are trained and optimized using the proposed combined loss function and their performances are evaluated in a cross-institutional test setting using several metrics such as mean absolute error (MAE), peak signal-to-noise ratio (PSNR), structural similarity index measure (SSIM), and multiscale structural similarity measure (MS–SSIM). The best-performing model (ResNet–BS) (PSNR = 34.0678; MS–SSIM = 0.9828) is used to suppress bones in the publicly available Shenzhen and Montgomery TB CXR collections. A VGG-16 model is pretrained on a large collection of publicly available CXRs. The CXR-pretrained model is then fine-tuned individually on the non-bone-suppressed and bone-suppressed CXRs of Shenzhen and Montgomery TB CXR collections to classify them as showing normal lungs or TB manifestations. The performances of these models are compared using several performance metrics such as accuracy, the area under the curve (AUC), sensitivity, specificity, precision, F-score, and Matthews correlation coefficient (MCC), analyzed for statistical significance, and their predictions are qualitatively interpreted through class-selective relevance maps (CRMs). It is observed that the models trained on bone-suppressed CXRs (Shenzhen: AUC = 0.9535 ± 0.0186; Montgomery: AUC = 0.9635 ± 0.0106) significantly outperformed (*p* < 0.05) the models trained on the non-bone-suppressed CXRs (Shenzhen: AUC = 0.8991 ± 0.0268; Montgomery: AUC = 0.8567 ± 0.0870).. Models trained on bone-suppressed CXRs improved detection of TB-consistent findings and resulted in compact clustering of the data points in the feature space signifying that bone suppression improved the model sensitivity toward TB classification.

## 1. Introduction

The World Health Organization (WHO) reports that millions of people suffer from lung-related diseases and their complications worldwide [1]. Chest X-rays (CXRs) are the most frequently performed diagnostic examination that helps detect various cardiopulmonary abnormalities. Portable digital CXRs are becoming part of modern point-of-care diagnostics for pulmonary abnormalities including tuberculosis (TB) [2]. However, it can be difficult for radiologists and computer-aided diagnostic (CADx) systems to detect and localize subtle findings related to TB when they occur in apical regions in which lung parenchyma is obscured by overlying ribs and the clavicles [3]. Due to the two-dimensional nature of image projection, the posterior and anterior bony structures on a typical CXR overlap with the lung tissues, resulting in a cross-hatching pattern. Further, the resulting strong edges from ribs and clavicles may occlude abnormalities in the lung regions thereby complicating diagnosis. Therefore, removing the superimposing bony structures could assist in reducing interpretation errors and enhance the value of screening digital chest radiography in underserved and remotely located populations [4].

Bone suppression involves subtracting the bones from the CXRs to create a soft-tissue image. It would be of potential use to radiologists and CADx systems in screening for subtle lung abnormalities by increasing the quality of soft tissue visibility. A common practice for suppressing bony structures involves the use of dual-energy subtraction (DES) chest radiography. The DES-based radiographic acquisition is performed to improve diagnosis by producing two different images, thereby separating the bony structures from the soft tissues. However, compared to conventional CXRs, DES has several limitations, namely, (a) DES radiography exposes the subject to slightly higher radiation doses compared to conventional CXR acquisition protocols and is not recommended for patients younger than 16 years of age [5]; (b) DES is not used for portable chest radiography, which limits its use in low and middle resource regions (LMRR); and (c) DES is performed only on the posterior-anterior view. 

Literature review reveals several image processing techniques for automated detection and removal of bony structures in CXRs [6,7]. In one study [8], the authors used a multiresolution artificial neural network to generate bony structures, subtracted these from the original CXRs to suppress the clavicles and ribs, and generated soft-tissue images. Another study [9] used independent component analysis to separate the ribs and soft tissues in CXRs to increase the visibility of lung nodules. Following this study, subsequent research adopted bone suppression to improve the detection of lung nodules and other pulmonary abnormalities [10,11,12], including pneumonia detection [13]. 

Inspired by their superior performance in natural and medical image recognition tasks [14,15], convolutional neural networks (ConvNets) have supplanted traditional techniques to perform bone suppression in CXRs. In one study [16], the authors used a cascade of ConvNets to predict bony structures at multiple resolutions and fused them to produce the final estimate of the bone image. The fused images are subtracted from their respective CXRs to produce soft-tissue images. In another study [17], the authors used a custom ConvNet model to classify the original, lung-segmented, and bone-suppressed versions of the Japanese Society of Radiological Technology (JSRT) CXR dataset [18]. It was observed that the model trained on the bone-suppressed dataset offered superior performance toward nodule detection, compared to those trained on the original and lung-segmented datasets. 

Bone suppression would help detect TB-consistent findings that often manifest in the apical lung regions so that their visibility is not obstructed by the occlusion of ribs and clavicles [3,19]. The effect of bone suppression on improving TB detection is discussed in the literature. For instance, the authors [20] fused information from local and global texture descriptors and a clavicle detection system toward detecting TB manifestations in CXRs. The performance with the fused detection system was observed to be superior (area under the curve (AUC) = 0.86), compared to using only the textural features (AUC = 0.67). In another study [21], the authors compared the performance of two CADx systems toward detecting TB-consistent findings in CXRs. One of the systems was trained on bone-suppressed images generated by commercially available software and the other was trained using original CXRs. It was observed that the CADx system trained on bone-suppressed images delivered superior performance in classifying CXRs as showing TB-consistent findings or normal lungs, compared to the other CADx system trained on the original CXRs. CXRs were digitally reconstructed from CT images in another study [22]. The authors suppressed bones in these reconstructed CXRs by leveraging a bone decomposition model that was trained on unpaired CT images. A ConvNet-based model was proposed [23] to extract bones from CXRs and subtract them from the original input CXRs to generate bone-subtracted images. In another study [24], the authors performed multilevel wavelet-based decomposition to predict bone images and subtract them from the original CXRs to produce bone-suppressed images. Other than these studies, literature that discusses the effect of bone suppression on TB detection is limited. Additionally, these methods involve multiple steps including predicting bony structures and then subtracting them from the original image to create bone-suppressed images. However, the literature is limited considering the availability of a bone suppression approach that would directly produce a bone-suppressed image from the input CXR. At the time of writing this manuscript, there is no literature available that evaluates the use of ConvNet-based bone suppression models toward improving automated detection of TB-consistent findings in CXRs. 

In this study, we propose a systematic methodology toward training customized ConvNet-based bone suppression models and evaluating their performance toward classifying and detecting TB-consistent findings in CXRs: First, we retrain an ImageNet-trained VGG-16 [25] model on a large-scale collection of publicly available CXRs from varied sources, where images were acquired for different clinical goals, to help it learn CXR modality-specific features and classify them as showing normal lungs or other pulmonary abnormalities. This model is hereafter referred to as the CXR–VGG-16 model. We use the VGG-16 model since it has demonstrated superior classification and localization performances in CXR classification tasks [26]. Next, we assess the performance of the CXR–VGG-16 model toward classifying CXRs in the Shenzhen and Montgomery TB CXR collections [27] as showing normal lungs or pulmonary TB manifestations. These are referred to as the baseline models. Then, we train several customized ConvNet-based bone suppression models with varying architecture on the JSRT CXR dataset [18] and its bone-suppressed counterpart [28]. We conduct cross-institutional testing using the National Institutes of Health (NIH) clinical center (CC) dual-energy subtraction (DES) CXR test set [29]. The best performing model is then used to suppress the bones in the Shenzhen and Montgomery TB CXR collections. The CXR–VGG-16 model is individually fine-tuned on the bone-suppressed images of the Shenzhen and Montgomery TB CXR collections toward classifying them as showing normal lungs or pulmonary TB manifestations. They are referred to as bone-suppressed models. Finally, the performance of the baseline and bone-suppressed models is quantitatively compared through several performance metrics and analyzed for statistically significant differences. Additionally, the predictions of the baseline and bone-suppressed models are qualitatively interpreted using class-selective relevance map (CRM)-based visualization [30].

The contributions of this retrospective study are highlighted as follows:

(i) This is the first study to propose and compare the performance of several customized ConvNet-based bone suppression models with a diversified architecture, including a sequential ConvNet model, an autoencoder (AE) model, a residual learning (RL) model, and a residual network (ResNet) model toward suppressing bones in CXRs.

(ii) This study performs rigorous empirical evaluations, statistical significance analysis, and qualitative evaluation of the bone suppression and classification models. 

(iii) The models proposed in this study are not limited to the task of CXR bone suppression but can potentially be extended to other image denoising problems.

The rest of the study is organized as follows: Section 2 discusses the datasets and methods used, Section 3 interprets the results, and Section 4 discusses and concludes this study.

## 2. Materials and Methods

The materials and methods are further divided into the following sub-sections: (i) datasets and their characteristics, (ii) bone suppression models, (iii) evaluating bone suppression models, (iv) histogram similarity assessment, (v) classification models, and (vi) statistical analysis.

### 2.1. Datasets and Their Characteristics

The following CXR collections are used in this study:

(i) JSRT CXR: The JSRT [18] released a set of 247 CXR images with and without lung nodules. The collection includes 154 images with a nodule, of which 100 nodules are malignant, 54 are benign, and 93 images are without nodules. 

(ii) NIH–CC–DES CXR: A set of 27 DES CXRs is acquired as a part of routine clinical care using the GE Discovery XR656 digital radiography system [29]. The DES images were taken at 120 and 133 Kilovoltage-peak (kVp) to, respectively, capture the soft-tissue images and bony structures. This dataset is used as the cross-institutional test set to evaluate the performance of the bone suppression models proposed in this study. 

(iii) Shenzhen TB CXR: This de-identified dataset contains 326 CXRs with normal lungs and 336 abnormal CXRs showing various TB manifestations [27]. The CXRs are collected from Shenzhen No.3 hospital in Shenzhen, China. It is exempted from institutional review board (IRB) review (OHSRP#5357) by the National Institutes of Health (NIH) Office of Human Research Protection Programs (OHSRP) and made publicly available by the National Library of Medicine (NLM). An equal number of normal and abnormal CXRs (*n* = 326) is used in this study.

(iv) Montgomery TB CXR: The CXR images and their associated radiology reports in this collection are acquired through the TB control program of the Department of Health and Human Services of Montgomery County, Maryland, USA [27]. The collection includes 58 CXRs showing TB-consistent findings and 80 CXRs with normal lungs. The CXRs are de-identified to ensure patient privacy and are made publicly available. An equal number of normal and abnormal CXRs (*n* = 58) is used in this study. 

(v) Radiological Society of North America (RSNA) CXR: A subset of the NIH CXR dataset [31] is curated by the RSNA [32] and made publicly available. The collection includes 17,833 frontal CXRs showing various lung abnormalities and 8851 CXRs showing normal lungs. 

(vi) Pediatric pneumonia CXR: A collection of 4273 CXRs, acquired from children 1 to 5 years of age, showing bacterial and viral pneumonia manifestations, and 1493 normal CXRs is publicly available [33].

Table 1 provides the demographic details of the datasets used in this study.

### 2.2. Bone Suppression Models

The researchers from the Budapest University of Technology and Economics used their in-house clavicle and rib–shadow removal algorithms to suppress the bones in the 247 JSRT CXRs and made the bone-suppressed soft-tissue images publicly available [28]. Affine transformations including rotations (−10 to 10 degrees), horizontal and vertical shifting (−5 to 5 pixels), horizontal mirroring, zooming, median, maximum, and minimum, and unsharp masking are used to generate 4500 image pairs from this initial set of CXRs and their bone-suppressed counterparts. The augmented images are resized to 256 × 256 spatial resolution. The image contrast is enhanced by saturating the bottom and top 1% of all image pixel values. The grayscale pixel values are then normalized. 

Several ConvNet-based bone suppression models with varying architecture are trained on this augmented dataset. We evaluated their performance with the cross-institutional NIH–CC–DES test set. During training, we allocated 10% of the training data for validation using a fixed seed. Four different model architectures are proposed toward the task of bone suppression in CXRs as follows: (a) Autoencoder (AE) model (AE–BS) where BS denotes bone suppression; (b) Sequential ConvNet model (ConvNet–BS); (c) Residual learning model (RL–BS); and (d) Residual network model (ResNet–BS). The architectures of these models are as follows:

(i) AE–BS Model: The AE–BS model is a convolutional denoising AE with symmetrical encoder and decoder layers. The encoder consists of three convolutional layers with 16, 32, and 64 filters, respectively. The size of the input is decreased twice at the encoder layers and increased correspondingly in the decoder layers. As opposed to the conventional denoising AEs, the noise in the proposed AE–BS model represents the bony structures. The model trains on the original CXRs and their bone-suppressed counterparts to predict a bone-suppressed soft-tissue image. Figure 1 illustrates the architecture of the proposed AE–BS model.

(ii) ConvNet–BS model: The ConvNet–BS model is a sequential model consisting of seven convolutional layers having 16, 32, 64, 128, 256, 512, and 1 filter, respectively. Zero paddings are used to preserve the dimensions of the input image at all convolutional layers. Lasso regularization (L1) penalties are used at each convolutional layer to induce penalty on weights that seldom contribute to learning meaningful feature representations. This helps in improving model sparsity and generalizing to unseen data. The deepest convolutional layer with the sigmoidal activation produces the bone-suppressed soft-tissue image. Figure 2 illustrates the architecture of the proposed ConvNet–BS model.

(iii) RL–BS model: The architecture of the RL–BS model consists of eight convolutional layers having 8, 16, 32, 64, 128, 256, 512, and 1 filter, respectively. Zero paddings are used at all convolutional layers to preserve the dimensions of the input image. The RL–BS model learns the residual error between the predicted bone-suppressed image and its corresponding ground truth. The deepest convolutional slayer produces bone-suppressed images. Figure 3 shows the architecture of the proposed RL–BS model. The RL–BS model learns the residual error between the predictions and ground truth to produce bone-suppressed images.

(iv) ResNet–BS model: The architecture of the proposed ResNet–BS model is illustrated in Figure 4. The residual design utilizes shortcuts to skip over layers thereby eliminating learning convergence issues due to vanishing gradients. This facilitates reusing previous layer activations until the weights are updated in the adjacent layer. These shortcuts lead to improved convergence and optimization and help to construct deeper models.

Inspired by [34], ReLU activation layers are not used outside the residual blocks. This literature [34] also demonstrates that batch normalization leads to loss of information and reduces the range tractability of activations. Hence, the batch normalization layer and the final ReLU activation are removed from each ResNet block. A sequence of 16 ResNet blocks are used, each having 64 filters of size 3 × 3 and zero paddings to preserve original image dimensions. Scaling layers with a scaling factor of 0.1 are added after the deepest convolutional layer in each ResNet block to scale down the residuals before adding them back to the convolutional path [35]. The deepest convolutional layer with the sigmoidal activation predicts the bone-suppressed image.

### 2.3. Evaluating Bone Suppression Models

The bone suppression models are trained to suppress the bony structures in the CXRs and produce soft-tissue images. This can be treated as an image denoising problem in which the bones are considered noise. To obtain superior bone suppression results, we aim to reduce the error between the predicted bone-suppressed image and its ground truth and maximize the structural similarity. The selection of the loss function plays a prominent role in the bone suppression task. 

In this study, the performance of the proposed bone suppression models is evaluated through constructing a loss function that benefits from the combination of mean absolute error (MAE) and multiscale structural similarity index measure (MS–SSIM) losses, herein referred to as *combined loss*. Other pixel-based evaluation metrics used in this study include peak signal-to-noise ratio (PSNR) and structural similarity index measure (SSIM). The mean-squared error (MSE), also known as L2 loss, is a pixel loss measure that computes the sum of the squared distance between the predicted image and its ground truth. However, MSE does not interpret the quality of the predicted image. The MAE, otherwise called L1 loss, computes the sum of absolute differences between the ground truth and the predicted image. Studies in the literature reported that, unlike MSE, MAE provides a more natural measure of average error and is useful in performing intercomparisons of average model performance errors [36]. PSNR computes the peak signal-to-noise ratio between the predicted and ground truth images. This ratio is used to provide a quantitative assessment of the predicted image. A higher value for PSNR indicates a higher quality of prediction. SSIM provides a measure of similarity between the ground truth and predicted images. A previous study [37] reveals that SSIM provides a superior indication of prediction performance as it exemplifies human visual perception. The MS–SSIM measure is an extension of SSIM that computes structural similarity at various scales and combines them. Another study [38] reveals that MS–SSIM is an improved measure to use, compared to SSIM while characterizing the performance of the models because (i) it is measured over multiple scales, and (ii) it is demonstrated to preserve contrast at higher frequencies, compared to SSIM. On the other hand, MAE preserves luminance and contrast in the predicted image. The mathematical formulations of these metrics can be found in the literature [36,37,38]. We propose to train the bone suppression models using a combined loss function that benefits from both MAE and MS–SSIM as shown in equation [1].
(1)$${\mathrm{Loss}}_{\mathrm{Combined}}=\left(\Omega \times \mathrm{MS}-\mathrm{SSIM}\right)+\left(\left(1-\Omega \right)\times \mathrm{MAE}\right)\;$$

We set the value of Ω = 0.84 after empirical evaluations. Greater weight is given to MS–SSIM since we want the bone-suppressed image to be highly similar (i.e., least structural alteration) to the ground truth. The MAE is given lower significance in this measure since it focuses on overall luminance and contrast in the image, which is expected to change due to bone (white pixels) suppression.

### 2.4. Histogram Similarity Assessment

The histograms of the ground truth and the bone-suppressed image predicted by the proposed models are plotted and compared to observe their tonal distributions. Various metrics including correlation, intersection, chi-square distance, Bhattacharyya distance, and Earthmover distance (EMD) are used to compare these histograms and provide a measure of similarity. The higher the value of correlation and intersection, the closer (or more similar) is the histogram of the image pairs. This implies that the histogram of the predicted bone-suppressed image closely matches that of the ground truth. For distance-based metrics including chi-square, Bhattacharyya, and EMD, a smaller value indicates a superior match between the histogram pairs, signifying that the predicted bone-suppressed image closely matches that of the ground truth. The mathematical formulations of these metrics can be found in the literature [39].

### 2.5. Classification Models

In this study, an ImageNet-pretrained VGG-16 model [25] is retrained on a large collection of CXRs combined using RSNA CXR and pediatric pneumonia CXR data collections producing sufficient diversity in terms of image acquisition and patient demographics to learn the characteristics of abnormal and normal lungs. This VGG-16 model is truncated at its deepest convolutional layer and appended with a global average pooling (GAP) layer, a dropout layer with an empirically determined dropout ratio (0.5), and an output layer with two nodes to predict probabilities of the input CXRs as showing normal lungs or other pulmonary abnormalities. This CXR modality-specific retraining helps in improving the specificity of the network weights conforming to the CXR classification task under study. This approach is followed to learn CXR modality-specific characteristics about the normal lungs and an extensive selection of pulmonary abnormalities. The modality-specific knowledge would be relevant to be transferred to the CXR classification task, as compared to using the ImageNet weights from the natural image processing domain. A previous study [40] shows the benefits of using CXR modality-specific models retraining toward improving classification and localization performance and model generalization. During CXR modality-specific pretraining, the data are split at the patient level into 90% for training and 10% for testing. We allocated 10% of the training data for validation using a fixed seed value. This CXR–VGG-16 model is fine-tuned on the original Shenzhen and Montgomery TB CXR collection (baseline models) and their bone-suppressed counterparts (bone-suppressed models) to classify them as showing normal lungs or pulmonary TB manifestations. The bone-suppressed datasets are constructed by using the best-performing bone suppression model among the proposed models. For the finetuning task, fourfold cross-validation is performed in which the baseline and bone-suppressed CXRs in the Shenzhen and Montgomery TB collections are split at the patient level into four equal folds. The hyperparameters of the models are tuned while training on the three folds and validating with the fourth fold. The validation process is repeated with each fold, resulting in four different models.

During model training, data are augmented with random horizontal and vertical pixel shifts (−5 to 5 pixels), horizontal mirroring, and rotations (−10 to 10 degrees) to introduce data diversity into the training process and reduce overfitting to the training data. Class weights are used to penalize majority classes and reduce class imbalance errors. The models are trained and evaluated using stochastic gradient descent (SGD) optimization to estimate learning error and classification performance. Callbacks are used to store checkpoints of the models. The model weights delivering superior performance with their respective validation fold are used for further analysis.

The ground truth disease annotations for the Montgomery TB dataset were provided by an expert radiologist with more than 45 years of experience. The ground truth disease annotations for a subset (*n* = 68) of the Shenzhen TB dataset were provided by another expert radiologist with more than 30 years of experience. The web-based, VGG Image Annotator tool [41] was used by the radiologists to independently annotate the collections. The radiologists were asked to annotate TB-consistent ROIs using rectangular bounding boxes. These annotations were exported to JSON for subsequent analyses.

The performance of the models is quantitatively compared using the following metrics and analyzed for statistical significance: (a) Accuracy, (b) AUC, (c) Sensitivity, (d) Specificity, (e) Precision, (f) F-measure, and (g) Matthews correlation coefficient (MCC). The predictions of the best-performing models trained on the baseline and bone-suppressed data are interpreted through CRM-based visualization. A Windows^®^ system with Intel Xeon CPU and NVIDIA GeForce GTX 1070 graphics card and Keras DL framework with Tensorflow backend is used to train the models. The trained models and codes are available at https://github.com/sivaramakrishnan-rajaraman/CXR-bone-suppression.

### 2.6. Statistical Analysis

We performed statistical analyses to identify the existence of a statistically significant difference in performance between the models. For the bone suppression task, we used 95% confidence intervals (CI) as the “Wilson” score interval for the MS–SSIM metric to compare the performance of the proposed bone suppression models and estimate their precision through the error margin. For the classification task, we used one-way analysis of variance (ANOVA) [42] to investigate if there exists a statistically significant difference in the MCC values obtained using the baseline and bone-suppressed models. Before performing one-way ANOVA analysis, we conducted Shapiro–Wilk, and Levene analyses [43] to check if the assumptions of data normality and variance homogeneity are satisfied. We used R statistical software (v. 3.6.1) to perform these evaluations.

## 3. Results

Recall that the proposed bone suppression models are trained on the augmented JSRT dataset and its bone-suppressed counterpart. The performance of the trained models is evaluated with the cross-institutional NIH–CC–DES test set (*n* = 27). The performance achieved by the various bone suppression models is shown in Table 2.

It is observed that the 95% CI for the MS–SSIM metric achieved by the ResNet–BS model demonstrates a tighter error margin and hence higher precision, compared to the other models. The ResNet–BS model demonstrated the least values for the combined loss, MAE, and MS–SSIM_loss_ and superior values for PSNR, SSIM, and MS–SSIM. The ResNet–BS model statistically significantly outperformed the AE–BS model (*p* < 0.05) and the ConvNet–BS and RL–BS models for the PSNR metric (*p* < 0.05). For other metrics, the ResNet–BS model demonstrated superior performance than the CNN–BS, and RL–BS models. Figure 5 shows the final bone suppression images along with the original unsuppressed CXR from a normal CXR in the NIH–CC DES test set. All approaches appear to show substantial suppression of the bony structures in the apical regions. For differentiation among them, quantitative indices are needed. A quantitative comparison of the bone-suppressed CXR images in Figure 5 is provided by histogram similarity comparisons in Figure 6 and Table 3 that follow. Based on the comparison findings, the ResNet–BS model was used in subsequent analyses. 

Figure 6 shows several comparisons of the histogram of the images predicted using the bone suppression models and the histogram of the ground truth using the sample CXR from Figure 5. It is observed from Figure 6d that the histogram of the bone-suppressed image predicted by the ResNet–BS model closely matched the ground truth, compared to the histogram obtained with other models. We assessed the similarity of the histograms of the predicted images to the ground truth through several performance metrics including correlation, intersection, chi-square distance, Bhattacharyya distance, and EMD, as shown in Table 3. We used Open Source Computer Vision (OpenCV) library (v. 3.4) to perform these evaluations.

We observed from Table 3 that the similarity of the ground truth to itself resulted in a value of 0 for all the distance measures and a value of 1 for the correlation metric. This demonstrates a perfect match. Higher values for the correlation and intersection metrics computed using the GT–ResNet–BS histogram pair demonstrate that the histogram of the ResNet–BS-predicted bone-suppressed image closely matches that of the ground truth image. For distance-based metrics including chi-square, Bhattacharyya, and EMD, a smaller value indicates a superior match between the histogram pairs. This signifies that compared to other models, the bone-suppressed image predicted by the ResNet–BS closely matches that of the ground truth.

The best performing ResNet–BS model is further used to suppress bones in Shenzhen and Montgomery TB CXR collections. Figure 7 shows the bone-suppressed instances of a sample CXR from the Shenzhen and Montgomery CXR collections. It is observed that the ResNet–BS model generalized to the Shenzhen and Montgomery TB CXR collections that are not seen by the model during training or validation. The bone shadows are completely suppressed, and the resolution of the CXRs is preserved. 

Recall that for the classification task, the CXRs in the Shenzhen and Montgomery TB CXR collections are split at the patient level into four equal folds for performing cross-validation studies. The mean performance of the cross-validated models is given in Table 4. It is observed that the classification performance achieved with the bone-suppressed models using the Shenzhen and Montgomery TB CXR collections is superior, compared to the baseline models. The bone-suppressed models demonstrated superior values for all performance metrics. 

We performed one-way ANOVA to analyze the existence of a statistically significant difference in the AUC and MCC metrics achieved by the baseline and bone-suppressed models trained on the Shenzhen TB CXR collection. One-way ANOVA assumes normality of data and homogeneity of variances. For the AUC metric, we observed that the *p*-values for Shapiro–Wilk and Levene analyses are greater than 0.05 (Shapiro–Wilk (*p*) = 0.1022 and Levene (*p*) = 0.1206). This demonstrated that the assumptions of the data normality and variance homogeneity are satisfied. Through one-way ANOVA analysis, we observed that a statistically significant difference existed in the AUC values achieved by the baseline and bone-suppressed models (F(1, 6) = 5.943, *p* = 0.005). This underscored the fact that the AUC values obtained by the bone-suppressed models are significantly superior to those achieved by the baseline models. We performed similar analyses using the MCC metric. We observed from Shapiro–Wilk and Levene analyses that the assumptions of the normal distribution of data and variance homogeneity hold valid (Shapiro–Wilk (*p*) = 0.7780 and Levene (*p*) = 0.4268). We observed that there existed a statistically significant difference in the MCC values obtained by the baseline and bone-suppressed models (F(1, 6) = 17.58, *p* = 0.00573). This demonstrated that the MCC values obtained by the bone-suppressed models are significantly higher compared to the baseline models.

A similar analysis is performed using the AUC and MCC metrics achieved by the cross-validated baseline and bone-suppressed models that are trained on the Montgomery TB CXR collection. Analyses of the AUC metric led to the observation that (i) the assumptions of data normality and variance homogeneity hold valid (Shapiro–Wilk (*p*) = 0.4102 and Levene (*p*) = 0.5510) and (ii) a statistically significant difference existed in the AUC values obtained by the baseline and bone-suppressed models (F(1, 6) = 11.13, *p* = 0.0157). Analyzing the MCC values led to the observation that (i) the assumptions of normal distribution of data and homogeneity of variances are satisfied (Shapiro–Wilk (*p*) = 0.6767 and Levene (*p*) = 0.808) and (ii) there existed a statistically significant difference in the MCC values obtained by the baseline and bone-suppressed models (F(1, 6) = 10.48, *p* = 0.0177). This underscored the fact that the AUC and MCC values obtained by the bone-suppressed models are significantly higher than the baseline models. These statistical evaluations demonstrated the fact that the classification performance achieved by the bone-suppressed models toward TB detection significantly outperformed those trained on non-bone-suppressed images. 

Figure 8 and Figure 9 show the following visualizations obtained using the best-performing cross-validated bone-suppressed model, respectively, using the Shenzhen and Montgomery TB CXR collection: (a) confusion matrix; (b) AUC–ROC curve; and (c) Normalized Sankey diagram. Recall that the bone-suppressed models demonstrated statistically superior values for all performance metrics, compared to their baseline counterparts.

We also used CRMs to interpret the predictions of the best-performing baseline and bone-suppressed models using the Shenzhen and Montgomery TB CXR collections to localize TB-consistent findings. Figure 10a,d show instances of original CXRs, respectively, from the Shenzhen and Montgomery TB CXR collections. The expert ground truth annotations are shown with red bounding boxes. Figure 10b,e show how the best-performing baseline models interpret their prediction toward localizing TB-consistent ROI. Figure 10c,f show the TB-consistent ROI localized by the best-performing bone-suppressed models. It is observed that the bone-suppressed models demonstrated superior TB-consistent ROI localization, compared to the baseline models. From Figure 10b,e, it is observed that the baseline models are learning the surrounding context but not meaningful features. The TB-consistent ROI localization achieved by the bone-suppressed models conformed to the expert knowledge of the problem, as observed from Figure 10c,f, and showed that it learned meaningful, salient feature representations. 

We performed a systematic visualization of the learned features in the initial and deepest convolutional layers of the trained bone-suppressed models to interpret the features detected for a given CXR image. Figure 11 and Figure 12 show the features learned by the first 64 filters using the best-performing bone-suppressed models for a sample CXR from the Montgomery and Shenzhen TB collections, respectively. From Figure 11 and Figure 12, we observed that the filters in the first convolutional layer learned the edges, contours, orientations, and their combinations, specific to the input image. However, in the deepest convolutional layer, the filter activations were abstracted to encode class-specific information. Additionally, the activation sparsity increased with model depth. This demonstrated that deeper convolutional layers encode class-specific details, while the initial layers contain image-specific activations. The bone-suppressed model distilled the input to repeatedly transform it to encode only class-relevant information with increasing depth while filtering out irrelevant information specific to the visual characteristics of the input CXR image.

We further used the CRM algorithm and the best-performing bone-suppression models to visualize the overall pulmonary location of TB manifestations in Shenzhen and Montgomery TB CXR collections. The average CRMs for the two datasets are shown in Figure 13. The steps taken to generate the average CRMs independently for the Shenzhen and Montgomery TB collection are (i) The average of CRMs were computed for the TB class in each dataset; (ii) the average of the ground truth lung masks for the Montgomery [24] and Shenzhen TB CXR [44] collections were computed, and (iii) a bitwise-AND operation was performed using the average CRMs and the averaged lung masks to visualize the activations in the lung ROI. The average CRMs appeared quite interesting and showed that the Shenzhen TB-positive group had primarily upper lobe CXR abnormalities. The average CRM obtained using the Montgomery TB CXR collection also showed upper lung predominance as well as other zones.

We visualized the learned features from the Shenzhen and Montgomery TB CXR collections by the best-performing baseline and bone-suppressed models using t-SNE [45]. The t-SNE is a dimensionality reduction technique that helps to visualize the learned feature space by embedding high-dimensional images into low dimensions while maintaining the pairwise distances of the points. The 512-dimensional vector extracted from the GAP layer of the baseline and bone-suppressed models is plugged into t-SNE to visualize feature embeddings in the two-dimensional space. From Figure 14, it is observed that the feature space learned by the bone-suppressed models demonstrated a better and more compact clustering of the normal and TB class features. Such feature clustering facilitates markedly superior separation between the classes, as compared to the baseline models. This improved behavior is observed with the bone-suppressed models fine-tuned on both Shenzhen and Montgomery TB CXR collections.

## 4. Discussion

Observations made from this study include the need for (i) CXR modality-specific model pretraining, (ii) model customization suiting the problem, (iii) statistical validation, (iv) localization studies with expert annotations conforming to the problem, and (v) feature embedding visualization. 

CXR modality-specific pretraining: Previous studies reveal that compared to using ImageNet weights, CXR modality-specific model pretraining results in learning meaningful modality-specific features that can be transferred to improve performance in a relevant classification task [40,46]. We performed CXR modality-specific pretraining using a selection of various publicly available CXR data collections to introduce sufficient diversity into the training process in terms of acquisition methods, patient population, and other demographics, to help the models broadly learn significant features from CXRs showing normal lungs and other pulmonary abnormalities. The learned knowledge is transferred to improve convergence and performance in a relevant classification task to classify CXRs as showing normal or TB manifestations. This approach may have helped the DL models to distinguish salient radiological manifestations of normal lungs and TB-consistent findings.

Model customization: Residual networks are one of the most commonly used backbones for computer vision tasks including segmentation, classification, and object detection [47]. The use of residual blocks helps construct and train deeper models since they alleviate the problem of vanishing gradients. In this study, we explored the use of residual networks in the context of an image denoising problem where the bony structures in the CXRs are considered noise. Through empirical evaluations, we observed that the proposed ResNet–BS model outperformed other models by demonstrating superior values for the PSNR, SSIM, and MS–SSIM metrics. The bone-suppressed image predicted by the ResNet–BS model effectively suppressed the bony structures and the image appeared sharp while preserving the soft tissues, rendering it suitable for lung disease screening/diagnosis. 

Statistical validation: Studies in the literature that accomplish bone suppression in CXRs have not performed a quantitative assessment of the bone-suppressed images by comparing them to their respective ground truths [48]. Statistical analysis would help evaluate model performance based on quantitative measures and help distinguish between realistic and uncertain assumptions. In our study, we performed histogram-based similarity assessments using several performance metrics including correlation, intersection, and other distance measures including chi-square distance, Bhattacharyya distance, and EMD to statistically demonstrate the closeness of the predicted bone-suppressed images with the ground truth. This led to the observation that, unlike other proposed bone suppression models, the histogram of the bone-suppressed CXRs predicted by the ResNet–BS model closely matched their respective ground truth images. We performed statistical analysis using 95% CI as the “Wilson” score interval to investigate the existence of a statistically significant difference in performance between the bone suppression models. We also performed one-way ANOVA analyses to observe the existence of a statistically significant difference in the classification performance using the baseline and bone-suppressed models. To this end, we observed a statistically significant difference (*p* < 0.05) existed in the MCC values obtained using the baseline and bone-suppressed models toward classifying the CXRs in the Shenzhen and Montgomery TB CXR collections. This demonstrated that the bone-suppressed models that are trained and evaluated individually on the Shenzhen and Montgomery TB CXR collections statistically significantly outperformed their baseline counterparts. 

Localization studies: We observed from the CRM-based localization study that the bone-suppressed models learned meaningful feature representations conforming to the expert knowledge of the problem under study. On the other hand, the baseline models, though demonstrating good classification accuracy, revealed poor TB-consistent ROI localization. These models learned the surrounding context irrelevant to the problem to classify the CXRs to their respective classes. This led to an important observation that the model accuracy is not related to its disease-specific ROI localization ability.

The average CRMs obtained using the Shenzhen and Montgomery TB CXR datasets, collected from TB clinics in two different countries, showed upper lung predominance. These observations conform to the findings in the literature [49] that discusses that 58% of patients with sputum-positive TB had upper lobe infiltrates. Another study [50] demonstrated that reactivation TB was especially common in the posterior segment of the upper lobe and the superior segment of the lower lobe. On frontal CXRs, those segments can appear to be in the midzone. The improved CRM localization achieved using the bone-suppressed models could be attributed to the fact that the suppression of bones helped to detect TB-consistent findings that often manifest in the apical lung regions so that their visibility is not obstructed by the occlusion of ribs and clavicles, thereby increasing model sensitivity.

Feature embedding visualization: We visualized the feature space learned by the baseline and bone-suppressed models using the t-SNE dimensionality reduction algorithm that embeds the learned high-dimensional features into the 2D space. To this end, we observed that the bone-suppressed model demonstrated a compact clustering of the features learned for the TB and normal classes. The decision boundary between the normal and TB categories are well defined, showing that meaningful feature embeddings are learned by the bone-suppressed models.

Limitations: This study, however, suffers from the following limitations: (i) To train and validate the proposed bone-suppression models, we used limited data that may not encompass a wide range of bone structure variability. With the increased availability of bone-suppressed CXRs in frontal and lateral projections, it would be possible to train deeper architectures with sufficient data diversity to build confidence in the models and improve their generalization to real-world data, and (ii) this study does not empirically identify the best classification model but investigates the impact of bone suppression on improving the classification performance in different TB datasets and substantiates the need for bone suppression toward improving TB detection. The impact of this approach on patient triage and treatment planning can only be theorized. Deriving guidance for them is beyond the scope of this study. 

In sum, models trained on bone-suppressed CXRs improved detection of TB-consistent findings resulted in compact clustering of the data points in the feature space, signifying that bone suppression improved the model sensitivity toward TB classification. The models proposed in this study are not limited to improving TB detection. The results suggest that the proposed ResNet–BS bone suppression model could be extended to other CXR applications such as improved performance in detecting and differentiating lung nodules, pneumonia, COVID-19, and other pulmonary abnormalities. This could further enhance the utility of digital CXRs for the evaluation of pulmonary disorders for underserved patients in low-resource or remote locations. We believe our results will improve human visual interpretation of TB findings, as well as automated detection in AI-driven workflows.

## Figures and Tables

**Figure 1 diagnostics-11-00840-f001:**
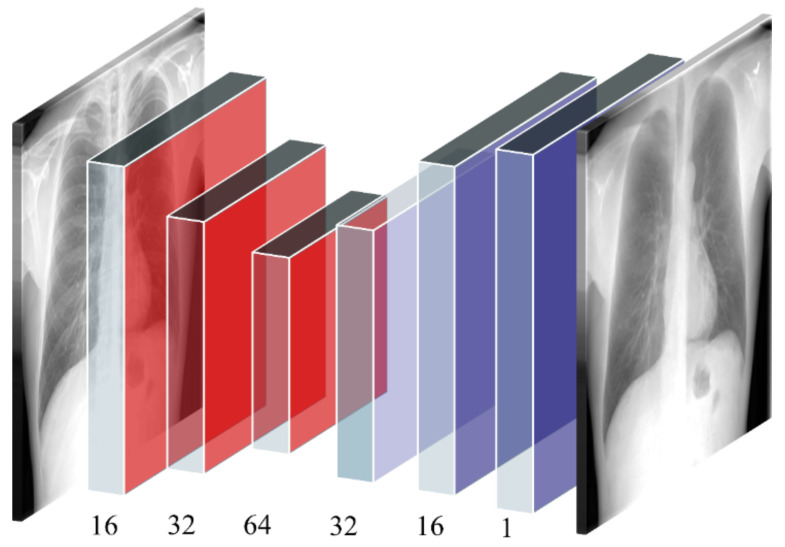
The architecture of the proposed AE–BS model. The AE–BS model has a symmetrical convolutional encoder (shown with red-colored boxes) and decoder (shown with blue-colored boxes) architecture.

**Figure 2 diagnostics-11-00840-f002:**
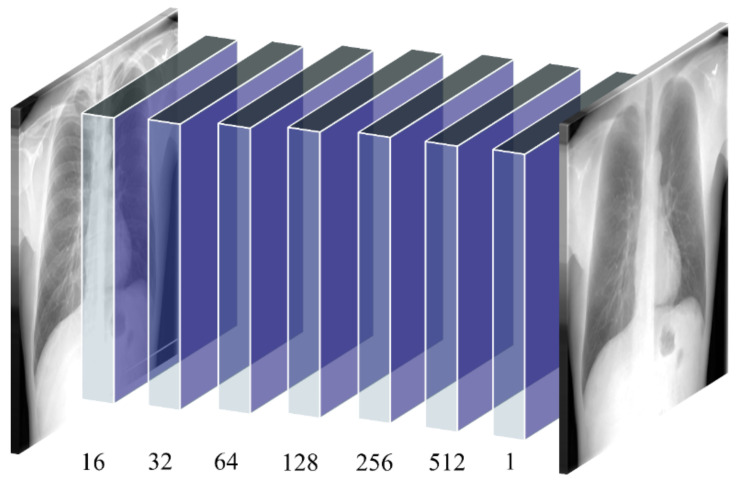
The architecture of the proposed ConvNet–BS model. The ConvNet–BS model has seven convolutional layers (shown with blue-colored boxes) with zero paddings to preserve original input dimensions. The deepest convolutional layer with the sigmoidal activation produces bone-suppressed soft-tissue images.

**Figure 3 diagnostics-11-00840-f003:**
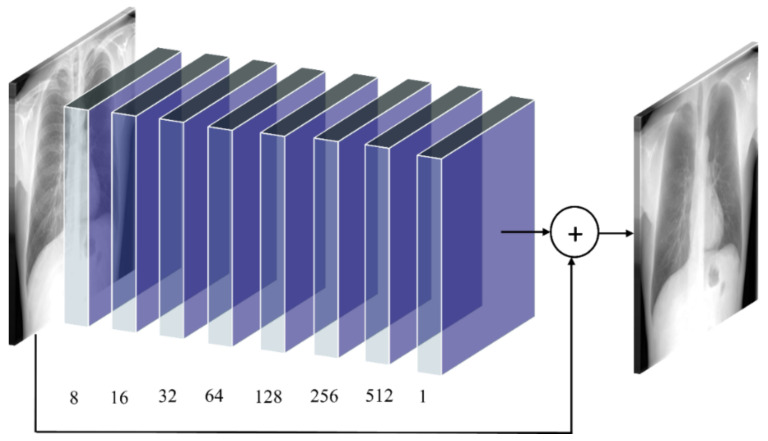
The architecture of the proposed RL–BS model.

**Figure 4 diagnostics-11-00840-f004:**
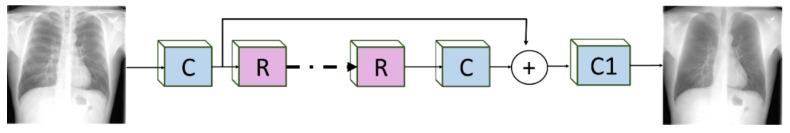
The architecture of the proposed ResNet–BS model. The convolutional block is denoted by C, having 64 filters of size 3 × 3 and zero paddings to preserve input dimensions. R denotes the modified ResNet block where the final ReLU activation is removed together with the batch normalization layer. The proposed model has 16 ResNet blocks. The deepest convolutional layer C1 with a single filter, zero paddings, and sigmoidal activation, predicts the bone-suppressed image.

**Figure 5 diagnostics-11-00840-f005:**
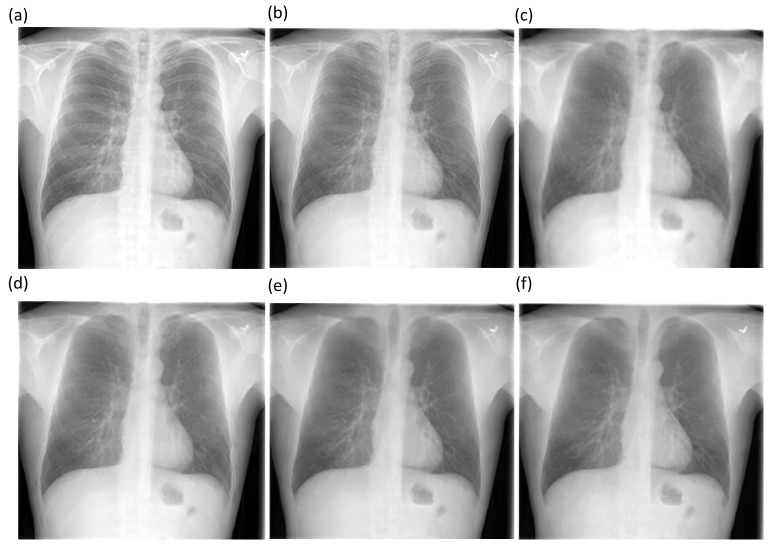
Bone-suppressed CXR images predicted by the proposed models using a CXR sample from the cross-institutional NIH–CC DES test set. (**a**) Original CXR; (**b**) AE–BS model; (**c**) ConvNet–BS model; (**d**) RL–BS model; (**e**) ResNet–BS model; and (**f**) Ground truth.

**Figure 6 diagnostics-11-00840-f006:**
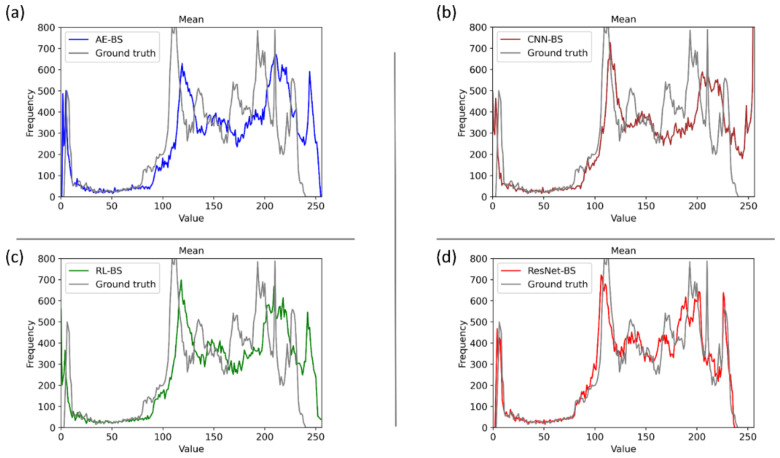
Comparing the histogram of the predicted image using the proposed bone suppression models and the ground truth using the sample CXR from Figure 2. (**a**) Ground truth and AE–BS model; (**b**) Ground truth and CNN–BS model; (**c**) Ground truth and RL–BS model; and (**d**) Ground truth and ResNet–BS model.

**Figure 7 diagnostics-11-00840-f007:**
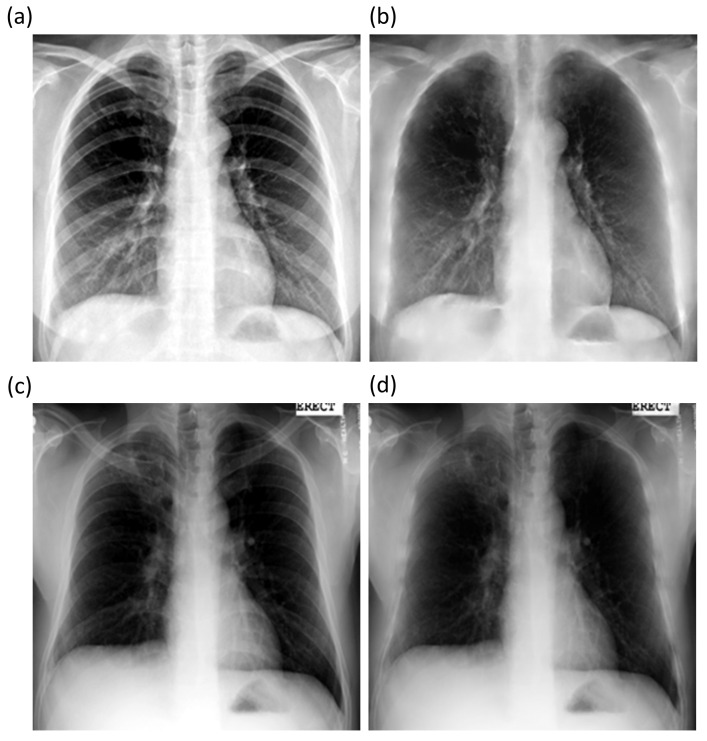
Bone-suppressed CXRs predicted by the ResNet–BS model using a sample CXR from the Shenzhen and Montgomery TB collection. (**a**) Shenzhen abnormal CXR; (**b**) Predicted bone-suppressed image; (**c**) Montgomery abnormal CXR; and (**d**) Predicted bone-suppressed image.

**Figure 8 diagnostics-11-00840-f008:**
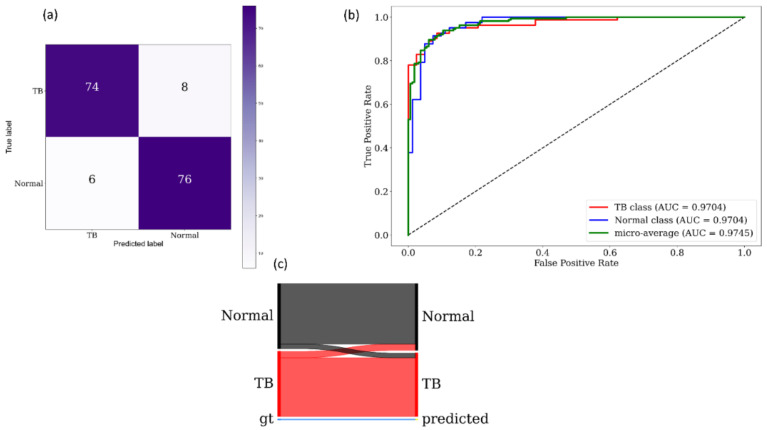
Performance visualization using the best-performing cross-validated bone-suppressed model that is trained and evaluated on the Shenzhen TB CXR collection. (**a**) Confusion matrix; (**b**) AUC–ROC curves; and (**c**) Normalized Sankey flow diagram.

**Figure 9 diagnostics-11-00840-f009:**
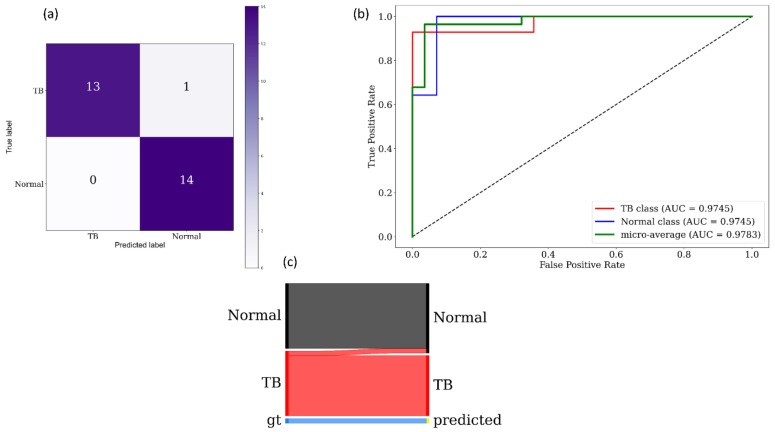
Performance visualization using the best-performing cross-validated bone-suppressed model that is trained and evaluated on the Montgomery TB CXR collection. (**a**) Confusion matrix; (**b**) AUC–ROC curves; and (**c**) Normalized Sankey flow diagram.

**Figure 10 diagnostics-11-00840-f010:**
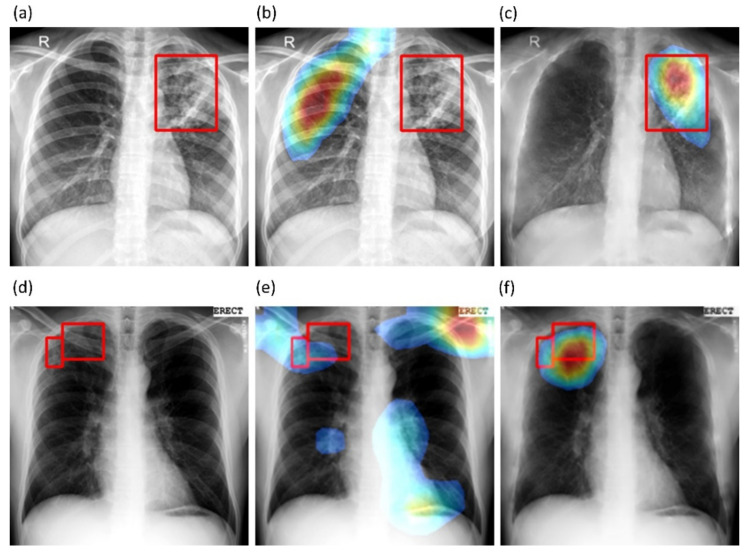
CRM-based TB-consistent ROI localization achieved by the best-performing baseline and bone-suppressed model, respectively, using a sample CXR from the Shenzhen and Montgomery TB CXR collection. (**a**) a CXR instance from the Shenzhen TB CXR collection with expert ground truth annotations (shown with red bounding boxes); (**b**) TB-consistent ROI localization achieved by the best-performing baseline model using the Shenzhen CXR instance; (**c**) TB-consistent ROI localization achieved by the best-performing bone-suppressed model using the Shenzhen CXR instance; (**d**) a CXR instance from the Montgomery TB CXR collection with expert ground truth annotations (shown with red bounding boxes); (**e**) TB-consistent ROI localization achieved by the best-performing baseline model using the Montgomery CXR instance, and (**f**) TB-consistent ROI localization achieved by the best-performing bone-suppressed model using the Montgomery CXR instance.

**Figure 11 diagnostics-11-00840-f011:**
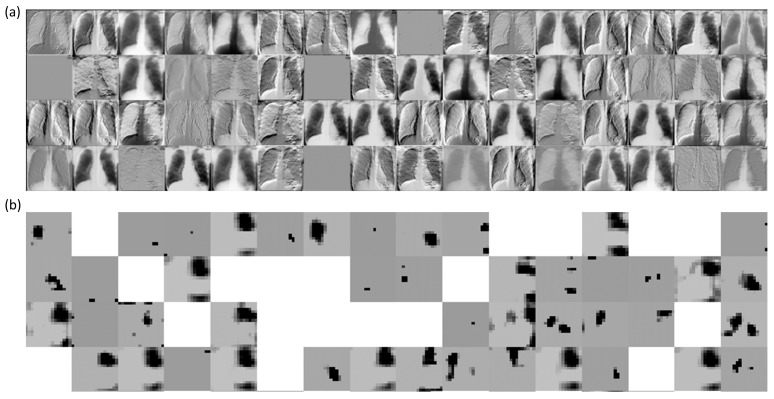
Visualizing the activations of the first 64 filters using the best-performing bone-suppressed model with a sample CXR from the Montgomery TB collection. The layer naming conventions follow Keras DL library. (**a**) block1-Conv1 layer and (**b**) block5-conv3 layer.

**Figure 12 diagnostics-11-00840-f012:**
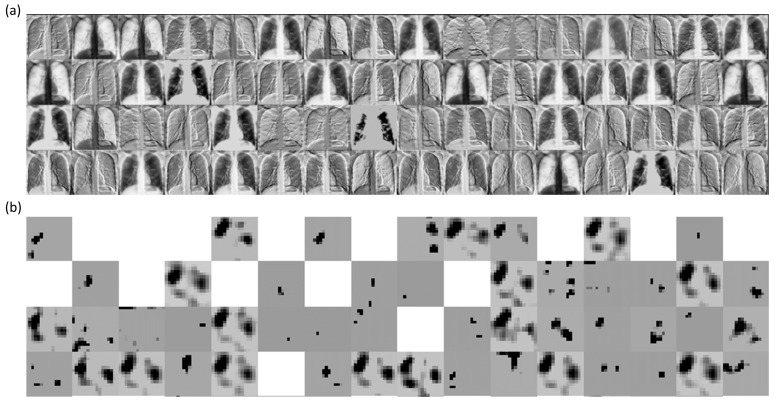
Visualizing the activations of the first 64 filters using the best-performing bone-suppressed model with a sample CXR from the Shenzhen TB collection. (**a**) block1-Conv1 layer and (**b**) block5-conv3 layer.

**Figure 13 diagnostics-11-00840-f013:**
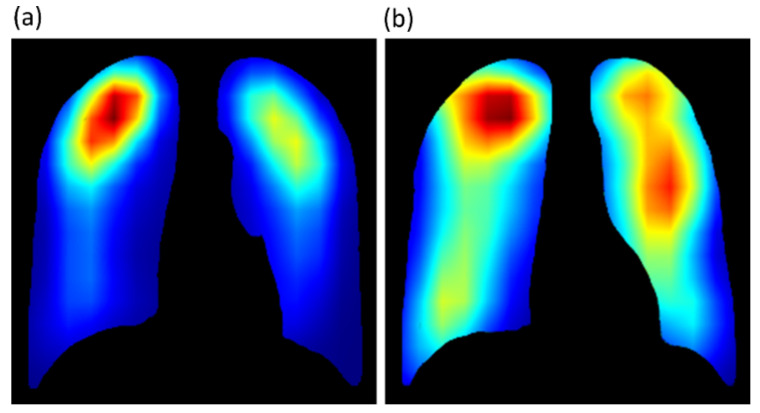
Average CRM computed for the TB class using the (**a**) Shenzhen and (**b**) Montgomery TB CXR collection.

**Figure 14 diagnostics-11-00840-f014:**
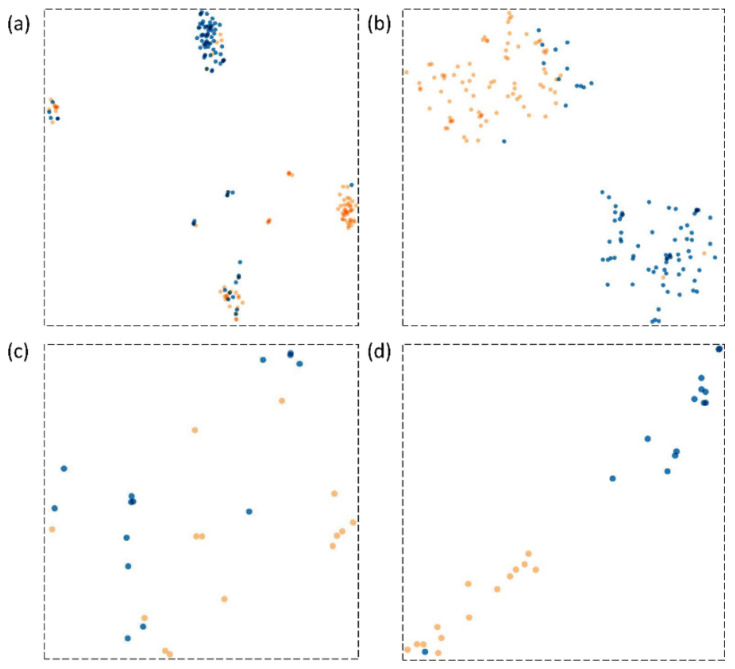
Feature embedding visualization with the t-SNE algorithm. The blue-colored points denote the feature embeddings for the TB class and the orange-colored points denote that of the normal class. (**a**) t-SNE visualization obtained by the best-performing baseline model evaluated using the Shenzhen TB CXR collection; (**b**) t-SNE visualization obtained by the best-performing bone-suppressed model evaluated using the Shenzhen TB CXR collection; (**c**) t-SNE visualization obtained by the best-performing baseline model evaluated using the Montgomery TB CXR collection, and (**d**) t-SNE visualization obtained by the best-performing bone-suppressed model evaluated using the Montgomery TB CXR collection.

**Table 1 diagnostics-11-00840-t001:** Demographic study. Details including patient count, sex, and the count of abnormal and normal images available for various datasets used in this study are shown. NA denotes Not Available. A total of 33,497 CXRs are included. Of these, 22,654 are abnormal with 394 being positive for TB (1.74% of abnormals, 1.18% of the entire sample).

Dataset	Total		Images	
	Male	Female	Normal	Abnormal
JSRT CXR	119	128	93	154
Pediatric pneumonia CXR	NA	NA	1493	4273
RSNA CXR	17,006	12,888	8851	17,833
Shenzhen TB CXR	449	213	326	336
Montgomery TB CXR	64	74	80	58

**Table 2 diagnostics-11-00840-t002:** Performance achieved by the proposed bone suppression models using the cross-institutional NIH–CC–DES test set. Data in parenthesis are 95% CI for the MS–SSIM values measured as the “Wilson” score interval. Combined loss = 0.16 * MAE + 0.84 * MS–SSIM_loss_. The best performances are denoted by bold numerical values in the corresponding columns. The ResNet–BS model statistically significantly outperformed the AE–BS model in all categories (*p* < 0.05), and the ConvNet–BS, and RL–BS models for the PSNR metric (*p* < 0.05). For other metrics, the ResNet–BS model demonstrated superior performance than the CNN–BS, and RL–BS models.

Model	Combined Loss	MAE	MS–SSIM_loss_	PSNR	SSIM	MS–SSIM
AE–BS	0.0251	0.0212	0.0258	30.462	0.9206	0.9742 (0.8759, 1.0)
ConvNet–BS	0.0217	0.0198	0.0221	30.9518	0.9352	0.9779 (0.8867, 1.0)
RL–BS	0.0211	0.0219	0.021	31.7495	0.9375	0.979 (0.8901, 1.0)
ResNet–BS	**0.0167**	**0.014**	**0.0172**	**34.0678**	**0.9492**	**0.9828 (0.9022, 1.0)**

**Table 3 diagnostics-11-00840-t003:** Histogram similarity assessment. The similarity of the histograms of the predicted images using the bone suppression models and their corresponding ground truths are measured. Bold numerical values denote superior performance in respective rows.

Method	Histogram Pairs				
	GT–GT	GT–AE–BS	GT–ConvNet–BS	GT–RL–BS	GT–ResNet–BS
Correlation	1	0.4368	0.4406	0.4644	**0.6723**
Intersection	10.6273	7.1058	7.1681	7.2151	**9.2880**
Chi-square distance	0	122.59	80.9075	60.30	**1.7931**
Bhattacharyya distance	0	0.4288	0.4272	0.4249	**0.3595**
EMD	0	0.0141	0.0135	0.0114	**0.0089**

**Table 4 diagnostics-11-00840-t004:** Mean performance achieved by the cross-validated models using the bone-suppressed and non-bone-suppressed (baseline) CXR instances of the Shenzhen and Montgomery TB CXR dataset. Acc = Accuracy; Sens. = Sensitivity; Spec. = Specificity; Prec. = Precision; F = F-measure. One-way ANOVA is performed using the MCC values obtained by the baseline and bone-suppressed cross-validated models to analyze for the existence of a statistically significant difference in performance. Bold numerical values denote superior performances in corresponding columns for the Shenzhen and Montgomery TB CXR collections. The performance of the bone-suppressed models is statistically significantly superior (*p* < 0.05) to the baseline models in all categories.

Dataset	Model	ACC	AUC	Sens.	Spec.	Prec.	F	MCC
Shenzhen (*n* = 326)	Bone suppressed	**0.8879 ± 0.0247**	**0.9535 ± 0.0186**	**0.8805 ± 0.0205**	**0.8954 ± 0.0423**	**0.8949 ± 0.0376**	**0.8873 ± 0.0233**	**0.7765 ± 0.0492**
	Baseline	0.8304 ± 0.0117	0.8991 ± 0.0268	0.8068 ± 0.0203	0.8537 ± 0.0345	0.8469 ± 0.0265	0.8259 ± 0.0089	0.6620 ± 0.0238
Montgomery (*n* = 58)	Bone suppressed	**0.9230 ± 0.0312**	**0.9635 ± 0.0106**	**0.8772 ± 0.0708**	**0.9687 ± 0.0625**	**0.9706 ± 0.0588**	**0.9188 ± 0.0345**	**0.8539 ± 0.0581**
	Baseline	0.7701 ± 0.0820	0.8567 ± 0.0870	0.7991 ± 0.1931	0.7411 ± 0.0342	0.7517 ± 0.0274	0.7682 ± 0.1039	0.5537 ± 0.1761

## Data Availability

All data supporting the findings of this study are publicly available and are cited in the manuscript. The NIH–CC DES dataset is available upon request.

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
