# Peer review of "Chest X-ray Bone Suppression for Improving Classification of Tuberculosis-Consistent Findings"

_diagnostics, 2021, doi:10.3390/diagnostics11050840_

Round 1

Reviewer 1 Report

Major points 

-    This paper is disclosed as preprint. See https://arxiv.org/ftp/arxiv/papers/2104/2104.04518.pdf

-    “We believe our results will improve human visual interpretation of TB findings, as well as automated detection in AI-driven workflows.” In visual evaluation and/or radiologists’ diagnosis, usefulness of authors’ model is not evaluated in this study. 

-    Total numbers of patients in confusion matrices of Figures 10 and 11 are wrong. Authors experiments may have flaws.  

-    “(c) DES is performed only on the posterior-anterior view.” This is also the limitation of this paper; this paper used and evaluated CXR of only posterior-anterior view. Please clarify the limitation in this paper.  

-    “(iii) The models proposed in this study are not limited to the task of CXR bone suppression but can potentially be extended to other image denoising problems.“ UNET-like or RED_CNN models (used for CT denoising) should be trained and evaluated for bone suppression, in addition to the four networks. See the followings. 

https://github.com/SSinyu/RED-CNN
https://github.com/ni-chen/fbpconv
https://ieeexplore.ieee.org/document/7949028

-    Instead of statistical tests for difference of MCC values, please perform statistical tests (e.g., Delong test) for difference of AUC values. 

-    “This dataset is used as the cross-institutional test set to evaluate the performance of the bone suppression models proposed in this study.” “Toward this classification task, four-fold cross-validation is performed in which the CXRs in the Shenzhen and Montgomery collections are split at the patient level into four equal folds.” For bone suppression model, the model performance was evaluated using cross-institutional test set. The model performance of TB classification model should be evaluated in the same way. 

Minor points 

-    “Several thousand people die every year from lung-related diseases and their complications[1].” According to WHO (http://www.emro.who.int/health-topics/chronic-obstructive-pulmonary-disease-copd/index.html), 3 million people died of Chronic obstructive pulmonary disease (COPD). WHO predicts that COPD will become the third leading cause of death worldwide by 2030. 

-    “An equal number of normal and abnormal CXRs (n = 326) is used in this study” “An equal number of normal and abnormal CXRs (n = 58) is used in this study” How did authors select the CXRs from the whole datasets. 

-    “Table 1 provides the demographic details of the datasets used in this study.” This sentence should be separated from (vi)

-    Choice of DL loss should be validated, compared with MAE or MSE.  

-    How to obtain histogram? Target/locations of histogram is unclear in this paper.

Author Response

We thank the reviewer for the valuable time and effort in providing us constructive comments toward this study. To the best of our knowledge, we have addressed the concerns of the reviewer so that the revised manuscript is considered acceptable for publication.

Q1. This paper is disclosed as preprint. See https://arxiv.org/ftp/arxiv/papers/2104/2104.04518.pdf

Author response:  We thank the reviewer for this comment. We believe that MDPI Diagnostics accepts submissions that have previously been made available as preprints provided they have not undergone peer review. (https://www.mdpi.com/journal/diagnostics/instructions#preprints).

Q2.  “We believe our results will improve human visual interpretation of TB findings, as well as automated detection in AI-driven workflows.” In visual evaluation and/or radiologists’ diagnosis, usefulness of authors’ model is not evaluated in this study. 

Author response:  We thank the reviewer, and we agree. Our study doesn't intend to compare AI-driven approaches with radiologist interpretations. We do hope that our AI approach could guide a radiologist and thus improve performance but that will have to be investigated in a future study.

Q3. Total numbers of patients in confusion matrices of Figures 10 and 11 are wrong. Authors experiments may have flaws.  

Author response:  Figure 10 (Figure 8 in the revised manuscript) illustrates the performance achieved using the Shenzhen TB CXR collection. Figure 11 (Figure 9 in the revised manuscript) shows the performance measures achieved using the Montgomery TB CXR collection. Hence, the number of patients would be different.

Q4. “(c) DES is performed only on the posterior-anterior view.” This is also the limitation of this paper; this paper used and evaluated CXR of only posterior-anterior view. Please clarify the limitation in this paper.  

Author response:  We thank the reviewer for these insightful comments. DES is typically performed in the PA view [5]. At present, we do not have any publicly available collections of bone-suppressed lateral CXRs. However, CXRs that are taken in community settings and portable CXRs often don't include a lateral view. On the other hand, portable digital CXRs are becoming part of modern point-of-care diagnostics for TB [2]. We performed bone suppression on frontal (both posterior-anterior and anterior-posterior) CXRs that help improve detection of TB-consistent findings that often manifest in the apical lung regions so that their visibility is not obscured by the occlusion of ribs and clavicles. Our results show that through bone suppression, such at-risk populations for TB are distinguishable. The single view CXR is acknowledged in the ‘Discussion’ Section as a limitation that can be minimized with digital CXR techniques and computerized evaluation in community screening.

Q5. “(iii) The models proposed in this study are not limited to the task of CXR bone suppression but can potentially be extended to other image denoising problems.“ UNET-like or RED_CNN models (used for CT denoising) should be trained and evaluated for bone suppression, in addition to the four networks. See the followings. 

https://github.com/SSinyu/RED-CNN
https://github.com/ni-chen/fbpconv
https://ieeexplore.ieee.org/document/7949028

Author response:  We thank the reviewer for pointing us to some of the important literature on CT denoising. We would use these models in our future studies on CT denoising to compare with our proposed approach. However, in the revised manuscript, we have referred to these studies while discussing the potential use of CNNs for visual recognition tasks.  

Author response:  The following studies are cited in the revised manuscript:

  1. Jin, K.H.; McCann, M.T.; Froustey, E.; Unser, M. Deep Convolutional Neural Network for Inverse Problems in Imaging. IEEE Trans. Image Process. 2017, doi:10.1109/TIP.2017.2713099.
  2. Chen, H.; Zhang, Y.; Kalra, M.K.; Lin, F.; Chen, Y.; Liao, P.; Zhou, J.; Wang, G. Low-Dose CT with a residual encoder-decoder convolutional neural network. IEEE Trans. Med. Imaging 2017, doi:10.1109/TMI.2017.2715284.

Q6. Instead of statistical tests for difference of MCC values, please perform statistical tests (e.g., Delong test) for difference of AUC values. 

Author response:  We thank the reviewer for the insightful comments. We agree that the DeLong test is an important statistical measure that is used to evaluate if the AUC of Model A is significantly different from Model B (p < 0.05.) In this study, we performed four-fold cross-validation using the TB datasets. In this regard, we have AUC values for each cross-validated fold when training the baseline and bone-suppressed models. Therefore, we performed the one-way analysis of variance (ANOVA) to determine whether there are any statistically significant differences between the means of the AUC values of the baseline and bone-suppressed models. We ensured that the assumptions of data normality of variances homogeneity hold good (through Shapiro-Wilk and Levene Test) so that one-way ANOVA gives a valid result. Such an evaluation helped us to substantiate that the bone-suppressed models significantly outperformed the non-bone-suppressed models in terms of both AUC and MCC parameters. We have removed Figure 8 and Figure 9 to avoid repetitions since the results are already discussed in the text.

Author action: The following changes are made to the revised manuscript:

We performed one-way ANOVA to analyze the existence of a statistically significant difference in the AUC and MCC metrics achieved by the baseline and bone-suppressed models trained on the Shenzhen TB CXR collection. One-way ANOVA assumes normality of data and homogeneity of variances. For the AUC metric, we observed that the p-value for Shapiro-Wilk and Levene analyses is greater than 0.05 (Shapiro-Wilk (p) = 0.1022 and Levene (p) = 0.1206). This demonstrated that the assumptions of the data normality and variance homogeneity are satisfied. Through one-way ANOVA analysis, we observed that a statistically significant difference existed in the AUC values achieved by the baseline and bone-suppressed models (F(1, 6) = 5.943, p = 0.005). This underscored the fact that the AUC values obtained by the bone-suppressed models are significantly superior to those achieved by the baseline models. We performed similar analyses using the MCC metric. We observed from Shapiro-Wilk and Levene analyses that the assumptions of the normal distribution of data and variance homogeneity hold valid (Shapiro-Wilk (p) = 0.7780 and Levene (p) = 0.4268). We observed that there existed a statistically significant difference in the MCC values obtained by the baseline and bone-suppressed models (F(1, 6) = 17.58, p = 0.00573). This demonstrated that the MCC values obtained by the bone-suppressed models are significantly higher compared to the baseline models.

A similar analysis is performed using the AUC and MCC metrics achieved by the cross-validated baseline and bone-suppressed models that are trained on the Montgomery TB CXR collection. Analyses of the AUC metric led to the observation that (i) the assumptions of data normality and variance homogeneity hold valid (Shapiro-Wilk (p) = 0.4102 and Levene (p) = 0.5510), and (ii) a statistically significant difference existed in the AUC values obtained by the baseline and bone-suppressed models (F(1, 6) = 11.13, p = 0.0157). Analyzing the MCC values led to the observation that (i) the assumptions of normal distribution of data and homogeneity of variances are satisfied (Shapiro-Wilk (p) = 0.6767 and Levene (p) = 0.808), and (ii) there existed a statistically significant difference in the MCC values obtained by the baseline and bone-suppressed models (F(1, 6) = 10.48, p = 0.0177). This underscored the fact that the AUC and MCC values obtained by the bone-suppressed models are significantly higher than the baseline models. These statistical evaluations demonstrated the fact that the classification performance achieved by the bone-suppressed models toward TB detection significantly outperformed those trained on non-bone-suppressed images. 

Q7. “This dataset is used as the cross-institutional test set to evaluate the performance of the bone suppression models proposed in this study.” “Toward this classification task, four-fold cross-validation is performed in which the CXRs in the Shenzhen and Montgomery collections are split at the patient level into four equal folds.” For bone suppression model, the model performance was evaluated using cross-institutional test set. The model performance of TB classification model should be evaluated in the same way. 

 Author response:  We thank the reviewer for these suggestions. The primary focus of this study is not to empirically identify the best classification model but to study the impact of bone suppression on improving the classification performance in different TB datasets and substantiate the need for bone suppression toward improving TB detection. The study aims to identify the best performing bone-suppression model from the selection of proposed model architectures that would help to best suppress bones in real-world CXR datasets. In this regard, we performed cross-validation studies and cross-institutional testing for the bone suppression task to estimate the skill, robustness, and generalization of the model on unseen data. Selecting the best classification model through cross-institutional testing is beyond the scope of this study, however, we plan to perform such evaluations in our future studies.

Q8. Minor points: “Several thousand people die every year from lung-related diseases and their complications[1].” According to WHO (http://www.emro.who.int/health-topics/chronic-obstructive-pulmonary-disease-copd/index.html), 3 million people died of Chronic obstructive pulmonary disease (COPD). WHO predicts that COPD will become the third leading cause of death worldwide by 2030. 

Author response:  We thank the reviewer for pointing us to these valuable statistics. We have included these details in the revised manuscript.

Author action: The following points are included in the revised manuscript:

Several thousand people die every year from lung-related diseases and their complications [1]. Millions of people suffer from pulmonary disorders like chronic obstructive pulmonary disease (COPD) which is predicted by the World Health Organization (WHO) (http://www.emro.who.int/health-topics/chronic-obstructive-pulmonary-disease-copd/index.html) to be the third leading cause of mortality worldwide by 2030.

Q9. “An equal number of normal and abnormal CXRs (n = 326) is used in this study” “An equal number of normal and abnormal CXRs (n = 58) is used in this study” How did authors select the CXRs from the whole datasets. 

Author response:  Thanks for these comments. It is imperative to perform patient-specific undersampling/oversampling on occasions when confronted with an imbalanced dataset. In this study, we perform patient-specific undersampling where we randomly removed patient-specific samples from the majority class to consequently reduce the number of CXRs in the majority class in the transformed dataset.

Q10. “Table 1 provides the demographic details of the datasets used in this study.” This sentence should be separated from (vi).

Author response:  Agreed and modified.

Q11. Choice of DL loss should be validated, compared with MAE or MSE. 

Author response:  We thank the reviewer for these comments. MSE is a pixel loss measure that does not help interpret the quality of the predicted image. MAE, on the other hand, computes the sum of absolute differences between the ground truth and the predicted image, thereby provided a more natural measure of average error, and is useful in performing inter-comparisons of average model-performance errors. MS-SSIM is an improved measure to use compared to SSIM while characterizing the performance of the models because (a) it is measured over multiple scales, and (b) it is demonstrated to preserve contrast at higher frequencies compared to SSIM. On the other hand, MAE preserves luminance and contrast in the predicted image. In this regard, we proposed a novel loss function that combines MAE and MS-SSIM metrics to evaluate bone suppression models. We awarded a greater weight to MS-SSIM (0.84) since we preferred the bone suppressed image to be highly similar (i.e., least structural alteration) to the ground truth. The MAE is given lower significance in this measure as it focuses on overall luminance and contrast in the image which are expected to change due to bone (white pixels) suppression. As well, we observed that the best-performing ResNet-BS achieved superior performance in terms of all performance metrics (combined loss: 0.0167; MAE: 0.014; PSNR: 34.0678; SSIM: 0.9492; MS-SSIM: 0.9828) compared to other models proposed in this study. The ResNet-BS model significantly outperformed the AE-BS model in all metrics (p < 0.05) and the ConvNet-BS and RL-BS models for the PSNR metric (p < 0.05).

Q12.  How to obtain histogram? Target/locations of histogram is unclear in this paper.

Author response:  We thank the reviewer for these valuable suggestions. We regret the lack of clarity in the original submission. In this study, we have shown a comparison of the histogram of the image predicted using the proposed bone suppression models and its respective ground truth for a sample CXR from the NIH-DES-CC- test set. We used OpenCV to plot histogram using the following parameters: (a) the number of parameters to collect data (n-dims) = 1; that is, we count the intensity values of each pixel in the greyscale CXR image; (b) the number of subdivisions in each dim (bins) = 16; (c) the limits for the measured values (range) = [0, 255]. We used the “calcHist” function to calculate the histograms and draw their envelopes. Then we used the “comparatist” function to compute parameters such as correlation, intersection, chi-square distance, Bhattacharyya distance, and Earthmover distance (EMD) metrics to express the degree of match of the predicted and ground truth histograms.  The higher the value of correlation and intersection, the more similar is the histogram of the image pairs. For distance-based metrics including chi-square, Bhattacharyya, and EMD, a smaller value indicates a superior match between the histogram pairs, signifying that the predicted bone-suppressed image closely matches that of the ground truth.

Reviewer 2 Report

The manuscript is well written and easy to follow.

Methods and results are clearly presented.

No clinical flaws are present.

Conclusione are supporter by results.

Figures are well presented.

Author Response

We render our sincere thanks to the reviewer for providing constructive comments toward our study.

Reviewer 3 Report

The paper presents a good application with solid experimental study 

I would suggest the points to improve the paper

1- add the highest found results to abstract 

2- add the training parameters 

3- visualization of learned features by Deep models, at least for the first and last convolutional layers. 

4- In regard to open science, authors could make their implementations available through git or other repositories.

5- I would suggest in the introduction section to add a paragraph that presents the work in deep learning regarding the same topic plus the issues with these models which this paper solved. 

Author Response

We render our sincere thanks to the reviewer for the valuable time and insightful comments toward our study.

Q1. The paper presents a good application with solid experimental study. I would suggest the points to improve the paper: 1- add the highest found results to abstract. 2. Add the training parameters.

Author response:  Agreed. The abstract is modified to include statistical results and training parameters.

Author action:  The abstract is modified as given below:

Chest X-rays (CXRs) are the most commonly performed diagnostic examination to detect cardiopulmonary abnormalities. However, the presence of bony structures such as ribs and clavicles can obscure subtle abnormalities resulting in diagnostic errors. This study aims to build a deep learning (DL)-based bone suppression model that identifies and removes these occluding bony structures in frontal CXRs to assist in reducing errors in radiological interpretation, including DL workflows, related to detecting manifestations consistent with tuberculosis (TB). Several bone suppression models with various deep architectures are trained and optimized using the proposed combined loss function and their performances are evaluated in a cross-institutional test setting using several metrics such as mean absolute error (MAE), peak signal-to-noise ratio (PSNR), structural similarity index measure (SSIM), and multi-scale structural similarity measure (MS-SSIM). The best-performing model (ResNet-BS) (PSNR = 34.0678; MS-SSIM = 0.9828) is used to suppress bones in the Shenzhen and Montgomery TB CXR collections. A VGG-16 model is pretrained on a large collection of publicly available CXRs. The CXR-pretrained model is then fine-tuned individually on the non-bone-suppressed and bone-suppressed CXRs of Shenzhen and Montgomery TB CXR collections to classify them as showing normal lungs or TB manifestations. The performances of these models are compared using several performance metrics such as accuracy, the area under the curve (AUC), sensitivity, specificity, precision, F-score, and Matthews correlation coefficient (MCC), analyzed for statistical significance, and their predictions are qualitatively interpreted through class-selective relevance maps (CRMs). It is observed that the models trained on bone-suppressed CXRs (Shenzhen: AUC = 0.9535±0.0186; Montgomery: AUC = 0.9635±0.0106) significantly outperformed (p < 0.05) the models trained on the non-bone-suppressed CXRs (Shenzhen: AUC = 0.8991±0.0268; Montgomery: AUC = 0.8567±0.0870) using the Shenzhen and Montgomery TB collections. Models trained on bone-suppressed CXRs improved detection of TB-consistent findings and resulted in compact clustering of the data points in the feature space signifying that bone suppression improved the model sensitivity toward TB classification.

Q3. Visualization of learned features by Deep models, at least for the first and last convolutional layers. 

Author response:  Agreed. The learned features are visualized for the first and last layers using the models trained on the Shenzhen and Montgomery TB CXR collections.

Author action:  The following are included in the revised manuscript:

We performed a systematic visualization of the learned features in the initial and deepest convolutional layers of the bone-suppressed models to interpret the features detected for a given CXR image. Figure 11 and Figure 12 show the features learned by the first 64 filters using the best-performing bone-suppressed models for a sample CXR from the Montgomery and Shenzhen TB collections, respectively.  From Figure 11 and Figure 12, we observed that the filters in the first convolutional layer learned the edges, contours, orientations, and their combinations, specific to the input image. However, in the deepest convolutional layer, the filter activations were abstracted to encode class-specific information. As well, the activation sparsity increased with model depth. This demonstrated that deeper convolutional layers encode class-specific details while the initial layers contain image-specific activations. The bone-suppressed model distilled the input to repeatedly transform it to encode only class-relevant information with increasing depth while filtering out irrelevant information about the specific visual characteristics of the input CXR image.

Figure 11. Visualizing activations of the first 64 filters using the best-performing bone-suppressed model with a sample CXR from the Montgomery TB collection. (a) block1-Conv1 layer, and (b) block5-conv3 layer. 

Figure 12. Visualizing activations of the first 64 filters using the best-performing bone-suppressed model with a sample CXR from the Shenzhen TB collection. (a) block1-Conv1 layer, and (b) block5-conv3 layer. 

Q4- In regard to open science, authors could make their implementations available through git or other repositories.

Author response:  We thank the reviewer for this insightful suggestion. The codes will be made publicly available at https://github.com/sivaramakrishnan-rajaraman/CXR-bone-suppression upon manuscript acceptance.

Q5- I would suggest in the introduction section to add a paragraph that presents the work in deep learning regarding the same topic plus the issues with these models which this paper solved. 

Author response:  We thank the reviewer for these valuable suggestions. We have included literature that proposed (i) general bone suppression in chest X-rays and (ii) bone suppression toward improving TB detection. We observed that the methods proposed in the literature involve multiple steps including predicting bony structures and then subtracting them from the original image to create bone-suppressed images. However, the literature is limited considering the availability of a bone suppression approach that would directly produce a bone-suppressed image from the input CXR. At the time of writing this manuscript, we observed that no literature is available that evaluates the use of ConvNet-based bone suppression models toward improving automated detection of TB-consistent findings in CXRs. To the best of our knowledge, this is the first study to propose and compare the performance of several customized ConvNet-based bone suppression models toward suppressing bones in CXRs. The proposed approach could further enhance the utility of digital CXRs for the evaluation of pulmonary disorders. We believe our results will improve human visual interpretation of TB findings, as well as automated detection in AI-driven workflows.

Reviewer 4 Report

In this article different bone-suppression AI models are evaluated on a small dataset with various problem-specific metrics. The best-performing model is selected to remove bones which is considered as noise. It is demonstrated that the outcome bone-suppressed images lead to a better performance, of another convolutional network which distinguishes between normal lungs and lungs with tuberculosis findings, than raw images. It is well written and provides detailed evidence that bone-suppression is useful in the specific task. According to the authors, they are the first to evaluate models which produce directly bone-suppressed images. The limitation which is stated by the authors, is the small dataset to evaluate the bone-suppression. 107-109: What was the task used for retraining? 332-333. Is validation set used in training or is it a test set? The pipeline is evaluated separately in two different datasets. It might be interesting to train with one e.g. Montgomery and test with the other e.g. Shenzen. 627-628 It is stated that “We observed from the CRM-based localization study that the model accuracy is not related to its disease-specific ROI localization ability.” However, by Figure 12, it may be concluded that the model accuracy might be related to its disease-specific ROI localization ability.

Author Response

We render our sincere thanks to the reviewer for the valuable time and effort spent in this review and for providing us insightful and constructive comments toward our study.

In this article different bone-suppression AI models are evaluated on a small dataset with various problem-specific metrics. The best-performing model is selected to remove bones which is considered as noise. It is demonstrated that the outcome bone-suppressed images lead to a better performance, of another convolutional network which distinguishes between normal lungs and lungs with tuberculosis findings, than raw images. It is well written and provides detailed evidence that bone-suppression is useful in the specific task. According to the authors, they are the first to evaluate models which produce directly bone-suppressed images. The limitation which is stated by the authors, is the small dataset to evaluate the bone-suppression.

Q1. 107-109: What was the task used for retraining?

Author response: We regret the lack of clarity in this regard. The ImageNet-trained VGG-16 model was retrained on a large-scale collection of publicly available CXRs from varied sources to classify them as showing normal lungs or other pulmonary abnormalities.

Author action: We including the retraining task details as shown below:

In this study, we propose a systematic methodology toward training customized ConvNet-based bone suppression models and evaluating their performance toward classifying and detecting TB-consistent findings in CXRs: First, we retrain an ImageNet-trained VGG-16 [22] model on a large-scale collection of publicly available CXRs from varied sources, where images were acquired for different clinical goals, to help it learn CXR modality-specific features and classify them as showing normal lungs or other pulmonary abnormalities. This model is hereafter referred to as the CXR-VGG-16 model.

Q2. 332-333. Is validation set used in training or is it a test set?

Author response: We regret the lack of clarity in the initial submission. We performed CXR modality-specific model pretraining, followed by finetuning to detect TB manifestations. During CXR modality-specific pretraining, the data are split at the patient level into 90% for training and 10% for testing. We allocated 10% of the training data for validation using a fixed seed value. The model is trained to classify CXRs as showing normal lungs or other pulmonary abnormalities. This CXR-pretrained VGG-16 model is then fine-tuned on the Shenzhen and Montgomery TB CXR collections (baseline models) and their bone-suppressed counterparts (bone-suppressed models) to classify them as showing normal lungs or pulmonary TB manifestations. During the fine-tuning step, we performed a four-fold cross-validation study due to limited data availability to ensure preventing model overfitting. Here, the baseline and bone-suppressed CXRs in the Shenzhen and Montgomery TB collections are split at the patient level into four equal folds. The hyperparameters of the models are tuned while training on the three folds and validating with the fourth fold. The validation process is repeated with each fold, resulting in four different models. We have included these details in the revised manuscript.

Author action: The following changes are included in the revised manuscript:

During CXR modality-specific pretraining, the data are split at the patient level into 90% for training and 10% for testing. We allocated 10% of the training data for validation using a fixed seed value. This CXR-VGG-16 model is fine-tuned on the original Shenzhen and Montgomery TB CXR collection (baseline models) and their bone-suppressed counterparts (bone-suppressed models) to classify them as showing normal lungs or pulmonary TB manifestations. For the finetuning task, four-fold cross-validation is performed in which the baseline and bone-suppressed CXRs in the Shenzhen and Montgomery TB collections are split at the patient level into four equal folds. The hyperparameters of the models are tuned while training on the three folds and validating with the fourth fold. The validation process is repeated with each fold, resulting in four different models.

Q3. The pipeline is evaluated separately in two different datasets. It might be interesting to train with one e.g. Montgomery and test with the other e.g. Shenzen.

Author response: We thank the reviewer for these suggestions. The primary focus of this study is not to empirically identify the best classification model but to study the impact of bone suppression on improving the classification performance in different TB datasets to substantiate the need for bone suppression toward improving TB detection. The study aims to identify the best performing bone-suppression model from the selection of proposed model architectures that would help to best suppress bones in real-world CXR datasets. In this regard, we performed cross-validation studies and cross-institutional testing for the bone suppression task to estimate the skill, robustness, and generalization of the bone suppression model on unseen data. Selecting the best classification model through cross-institutional testing is beyond the scope of this study, however, we aim to perform such evaluations in our future studies.

Q4. 627-628 It is stated that “We observed from the CRM-based localization study that the model accuracy is not related to its disease-specific ROI localization ability.” However, by Figure 12, it may be concluded that the model accuracy might be related to its disease-specific ROI localization ability.

Author response: We regret the lack of clarity in this regard. We observed that the baseline, non-bone-suppressed models, though demonstrating good classification accuracy, learned the surrounding context irrelevant to the problem to classify the CXRs to their respective classes. In this regard, we mentioned that the model accuracy is not related to its disease-specific ROI localization ability. We rephrased this sentence in the revised manuscript to convey clarity.

Author action: The following changes are included in the revised manuscript:

We observed from the CRM-based localization study that the bone-suppressed models learned meaningful feature representations conforming to the expert knowledge of the problem under study. On the other hand, the baseline models, though demonstrating good classification accuracy, revealed poor TB-consistent ROI localization. These models learned the surrounding context irrelevant to the problem to classify the CXRs to their respective classes. This led to an important observation that the model accuracy is not related to its disease-specific ROI localization ability.

Round 2

Reviewer 1 Report

  • “The validation process is repeated with each fold, resulting in four different models.” “Q9. “An equal number of normal and abnormal CXRs (n = 326) is used in this study” “An equal number of normal and abnormal CXRs (n = 58) is used in this study”” “Q3. Total numbers of patients in confusion matrices of Figures 10 and 11 are wrong. Authors experiments may have flaws. Author response: Figure 10 (Figure 8 in the revised manuscript) illustrates the performance achieved using the Shenzhen TB CXR collection. Figure 11 (Figure 9 in the revised manuscript) shows the performance measures achieved using the Montgomery TB CXR collection. Hence, the number of patients would be different.” If these sentences are right, total number of patients in confusion matrix of Figure 9 must be 326 and that of Figure 10 must be 56 (these CXR images should be processed with the 4 different models of CV). However, Figure 9 and 10 show that total numbers of patients in confusion matrices are 164 and 28. Again, I speculate that authors’ experiments may have flaws.

  • “Q2. “We believe our results will improve human visual interpretation of TB findings, as well as automated detection in AI-driven workflows.” In visual evaluation and/or radiologists’ diagnosis, usefulness of authors’ model is not evaluated in this study. Author response: We thank the reviewer, and we agree. Our study doesn't intend to compare AI-driven approaches with radiologist interpretations. We do hope that our AI approach could guide a radiologist and thus improve performance but that will have to be investigated in a future study.” In the paper, please clarify that your study doesn't intend to compare AI-driven approaches with radiologist interpretations.

  • “Q4. “(c) DES is performed only on the posterior-anterior view.” This is also the limitation of this paper; this paper used and evaluated CXR of only posterior-anterior view. Please clarify the limitation in this paper. Author response: We thank the reviewer for these insightful comments. DES is typically performed in the PA view [5]. At present, we do not have any publicly available collections of bone-suppressed lateral CXRs. However, CXRs that are taken in community settings and portable CXRs often don't include a lateral view. On the other hand, portable digital CXRs are becoming part of modern point-of-care diagnostics for TB [2]. We performed bone suppression on frontal (both posterior-anterior and anterior-posterior) CXRs that help improve detection of TB-consistent findings that often manifest in the apical lung regions so that their visibility is not obscured by the occlusion of ribs and clavicles. Our results show that through bone suppression, such at-risk populations for TB are distinguishable. The single view CXR is acknowledged in the ‘Discussion’ Section as a limitation that can be minimized with digital CXR techniques and computerized evaluation in community screening.” Again, I recommend that this paper show that your AI cannot handle side-view or etc.

  • Author reply. “The primary focus of this study is not to empirically identify the best classification model but to study the impact of bone suppression on improving the classification performance in different TB datasets and substantiate the need for bone suppression toward improving TB detection.” If so, please clarify this point in the limitation.

  • “Several thousand people die every year from lung-related diseases and their complications[1].” Purpose of my previous comment is that “Several thousand people” may be wrong based on WHO website.

  • Q12. How to obtain histogram? Target/locations of histogram is unclear in this paper. Author response: We thank the reviewer for these valuable suggestions. We regret the lack of clarity in the original submission. In this study, we have shown a comparison of the histogram of the image predicted using the proposed bone suppression models and its respective ground truth for a sample CXR from the NIH-DES-CC- test set. We used OpenCV to plot histogram using the following parameters: (a) the number of parameters to collect data (n-dims) = 1; that is, we count the intensity values of each pixel in the greyscale CXR image; (b) the number of subdivisions in each dim (bins) = 16; (c) the limits for the measured values (range) = [0, 255]. We used the “calcHist” function to calculate the histograms and draw their envelopes. Then we used the “comparatist” function to compute parameters such as correlation, intersection, chi-square distance, Bhattacharyya distance, and Earthmover distance (EMD) metrics to express the degree of match of the predicted and ground truth histograms. The higher the value of correlation and intersection, the more similar is the histogram of the image pairs. For distance-based metrics including chi-square, Bhattacharyya, and EMD, a smaller value indicates a superior match between the histogram pairs, signifying that the predicted bone-suppressed image closely matches that of the ground truth.” Authors’ response clarifies the way to obtain histogram. However, target/location of histogram is unclear in this response. From whole image?